# Cephalometric Screening Assessment for Superior Airway Space Narrowing—Added Value of Three-Dimensional Imaging

**DOI:** 10.3390/jcm13092685

**Published:** 2024-05-02

**Authors:** Axel Meisgeier, Florian Dürrschnabel, Simon Pienkohs, Annabell Weiser, Andreas Neff

**Affiliations:** 1Department of Oral and Craniomaxillofacial Surgery, UKGM GmbH, University Hospital Marburg, 35043 Marburg, Germany; duerrsch@med.uni-marburg.de (F.D.); simonpatrik.pienkohs@uk-gm.de (S.P.); weisera@students.uni-marburg.de (A.W.); neffa@med.uni-marburg.de (A.N.); 2Faculty of Medicine, Philipps University, 35043 Marburg, Germany

**Keywords:** airway obstruction, cephalometry, maxillofacial surgery, obstructive sleep apnea, orthodontics, orthognathic surgical procedures

## Abstract

**Background**: Assessing the morphology of the superior airway space is a crucial diagnostic step in the treatment planning of patients with obstructive sleep apnea syndrome (OSAS) or prior to orthognathic surgery. The aim of this study is to evaluate the necessary scope of a two-dimensional cephalometric assessment and the necessity of three-dimensional imaging in the identification of superior airway space narrowing (SASN). **Methods**: The computed tomography studies of 100 non-obese, non-OSAS patients were evaluated and analyzed retrospectively. Multiplanar reconstructions were created and underwent cephalometric evaluation. The three-dimensional superior airway morphology was segmented and measured for the minimal cross-sectional area (A_min_) and volume (V_0_). Patients were grouped according to A_min_ < 80 mm^2^ and V_0_ < 12 cm^3^. Cephalometric parameters (CPs) were analyzed according to A_min_ and V_0_ with an unpaired *t*-test, Pearson correlation, and ROC-curve analysis. **Results**: The CPs regarding sagittal airway space dimensions (IPAS, MPAS, SPAS) and mandibular body length (GoGn) show the strongest correlation to the three-dimensional minimal cross-sectional area (A_min_). The ROC-curve analysis classifying for SASN led to an AUC of 0.86 for IPAS, 0.87 for MPAS, 0.88 for SPAS, and 0.63 for GoGn. Three-dimensional imaging may further improve the diagnostic accuracy in the identification of SASN for IPAS below 13.5 mm, MPAS below 10.2 mm, SPAS below 12.5 mm, and GoGn below 90.2 mm. **Conclusions**: Two-dimensional cephalometric sagittal airway space diameters and mandibular body length are useful initial screening parameters in the identification of superior airway space narrowing. Nevertheless, as the correlation of two-dimensional cephalometric parameters with three-dimensional upper airway space narrowing is varying and highly dependent on acquisition circumstances, indications for three-dimensional imaging, if possible, in the supine position to evaluate upper airway space morphology should be provided generously, especially in patients with low but normal airway space parameters in two-dimensional cephalometry.

## 1. Introduction

Assessing the morphology of the superior airway space is a crucial diagnostic step in the surgical treatment planning of patients with obstructive sleep apnea, prior to combined orthodontic–orthognathic surgical treatment, as well as in aesthetic orthognathic surgery [1,2,3,4]. Cephalometric diagnostics are highly recommended by the current guidelines regarding sleep-related breathing disorders prior to surgical treatment [5,6,7]. Nevertheless, there is no clear recommendation regarding the necessary scope of such a cephalometric airway assessment or the indication for three-dimensional imaging [8].

Obstructive sleep apnea syndrome (OSAS) is the most common sleep-related breathing disorder characterized by repeated airway obstruction during sleep [5,6,7]. Anatomical abnormalities as well as non-anatomical factors, such as impaired muscle responsiveness, obesity, unstable respiratory control, and the low respiratory arousal threshold, are risk factors for OSAS [9]. OSAS patients are exposed to intermittent hypoxia, leading to poor sleep quality and increased risks of various systemic diseases like coronary heart disease, atrial fibrillation, arterial hypertension, stroke, and diabetes mellitus [10,11,12,13,14,15]. Therefore, its timely diagnosis and treatment is essential for the promotion of general health. The gold standard for OSAS diagnosis is standard polysomnography (PSG), increasingly supported by out-of-center systems for outpatient sleep monitoring [5,6,16,17]. The most common form of therapy for all severity levels of OSAS is positive airway pressure (PAP), in the form of continuous PAP (CPAP), as a long-term therapy [5,6]. The indications for the initiation of CPAP therapy result from the synopsis of clinical history, polysomnographic and instrumental findings, as well as existing concomitant diseases [18]. Several additional therapy options are established. Uvulopalatopharyngoplasty in combination with tonsillectomy (TE-UPPP), the stimulation of the hypoglossal nerve, orthognathic surgery with bimaxillary advancement, and tracheostomy as an ultima ratio are established surgical procedures with the highest levels of evidence [5,6,19,20,21,22]. Nocturnal positional therapy and a mandibular protrusion splint have been established as non-surgical therapy options also [23,24].

Due to its multifactorial emergence and the broad range of therapeutic options, OSAS is an interdisciplinary challenge, especially in patients where CPAP is not tolerated or not possible for other reasons [5,6,7]. In patients with an anatomically identifiable correlate of superior airway space narrowing (SASN) or a reduced superior airway space volume (RSAV), therapy using a bimaxillary osteotomy with maxillomandibular advancement may be offered. While there is no convincing indication for routine X-ray cephalometry or CT scans as a primary diagnostic tool in OSAS, it is undeniably important when it comes to providing sufficient information and advice regarding the risks and chances of maxillomandibular surgery, as it may provide insights into the anatomical parameters regarding airway morphology and the characterization of craniofacial structures which are not only determinate factors in the pathogenesis of certain OSAS cases, but surgically addressable objectives in OSAS therapy, as recommended by current guidelines such as the German S3 guideline “Sleep-Related Breathing Disorders” [5,25,26]. In addition, orthognathic surgery always influences the morphology of the superior airway space, which can contribute to an increasing SASN or even the induction of OSAS [1,4].

The two-dimensional cephalometric factors associated with SASN, RSAV, and OSAS are discussed in a highly controversial manner [26,27,28,29,30]. Moreover, there is little information available with regard to additional three-dimensional parameters or the correlation of two-dimensional parameters with three-dimensional airway morphology [31,32,33,34]. Two-dimensional cephalometric assessments seem to be only a moderate prognostic tool prior to surgery. About 15% of the patients treated with maxillomandibular advancement do not respond as expected [35]. An improvement in the preoperative assessment of patients is essential and clinically meaningful to avoid unnecessary therapy in non-responding patients prior to surgery, where additional three-dimensional assessments may offer additional information. The aim of this study is to evaluate the necessary scope of such a cephalometric assessment regarding the indication for three-dimensional imaging.

## 2. Materials and Methods

### 2.1. Patients

This investigation was designed as a retrospective assessment. The study protocol was approved by the Institutional Ethics Board at Phillips-University Marburg, Germany (23/198 RS; 9 August 2023). Planning and execution followed the rules of the declaration of Helsinki. The study population comprised 100 non-obese, non-OSAS patients aged 18 years or older, out of the continuous spectrum of sagittal and vertical discrepancies of the facial skeleton prior to orthognathic surgery between January 2016 and December 2021. Patients with a history of head and neck trauma, cleft lips and palates, craniofacial syndromes, or major surgery in the orofacial region were excluded.

### 2.2. Computed Tomography

All patients were assessed using computed tomography using a Somatom Definition AS Dual Source CT scanner (Siemens Healthineers, Erlangen, Germany) with a slice thickness of 1 mm. All CT scans were taken with the subjects in the supine position with the habitual occlusion and lips at rest. Multiplanar reconstructions (lateral, frontal, and panoramic cephalometric X-ray) were created using Dolphin 3D 11.9 software (Dolphin Imaging, Chatsworth, CA, USA).

### 2.3. Cephalometric Assessment

Cephalometric assessment and airway segmentation was performed using Dolphin 3D 11.9 software by one observer (A.M.). To confirm the reproducibility and reliability of our results, inter- and intraobserver bias was tested in a randomly selected subgroup of 20 patients from the study sample by two separate investigators (A.M. and F.D.), using the intraclass correlation coefficient (ICC, absolute-agreement, 2-way mixed-effects model). According to Koo at al., the ICC was reported as poor for ICC < 0.50, moderate for ICC = 0.50 to <0.75, good for ICC 0.75 to <0.9, and very good for ICC ≥ 0.90 [36]. Thirty-eight cephalometric parameters (CPs) for skeletal and soft tissue, including the cranial base, face height, maxilla and mandible, soft palate, hyoid, and upper airway, were measured according to their definition shown in Table 1. The reconstruction of the lateral cephalometric X-ray and superior airway space segmentation is shown in Figure 1.

### 2.4. Data Analysis

During the descriptive phase, data were expressed as mean ± standard deviation when normally distributed, or as median ± interquartile range when not normally distributed. Normal distribution was tested with the Kolmogorov–Smirnov test. The patients were divided into four groups according to the identification of a three-dimensional upper airway narrowing, defined as a minimal cross-sectional area of less than 80 mm^2^ (Group A1: A_min_ < 80 mm^2^; Group A2: A_min_ ≥ 80 mm^2^) and reduced upper airway space volume of less than 12 cm^3^ (Group B1: V_PAS_ < 12 cm^3^; Group B2: V_PAS_ ≥ 12 cm^3^) [37]. An unpaired *t*-test was used for intergroup comparison. Gender ratios were compared with the chi-squared test. Correlations were calculated using the Pearson correlation coefficient. According to Hemphill et al., the correlation was reported weak r < 0.20; medium r = 0.20 to 0.30; and strong r > 0.30 [38]. The correction for multiple testing was performed with a false discovery rate adjustment according to Benjamini and Hochberg [39]. Multiple linear regression with stepwise inclusion (F < 0.05) and exclusion (F > 0.10) was performed to identify factors independently associated with three-dimensional upper airway space narrowing and reduced upper airway space volume. Individual parameters and the regression models were analyzed using the receiver operating characteristic (ROC) curve and the area under the curve (AUC) to obtain cut-off values for the identification of three-dimensional SASN and RSAV. For all tests, a significance level of 5% was used. To improve the accuracy when compared to the two-way, single cut-off classification, an intermediate range of values was defined as an indication for three-dimensional imaging. The lower cut-off value was defined as the individual value where the sensitivity in the ROC-curve analysis is 90%. The upper cut-off value was defined as the individual value where the specificity in the ROC-curve analysis is 90%, leading to an overall prevalence independent accuracy of >90%. Statistical analysis was performed using IBM SPSS Statistics Version 29.0 (SPSS GmbH, Munich, Germany).

## 3. Results

### 3.1. Demographic Characteristics

A total of 100 patients were included in the study (53 females, 47 males), with a mean age of 27.2 ± 8.9 years (range 18–55 years). The patients were independently divided into the following subgroups: (A) according to the identification of a three-dimensional SASN (Group A1: SASN A_min_ < 80 mm^2^, Group A2: No SASN A_min_ ≥ 80 mm^2^), and (B) the identification of an RSAV (Group B1: RSAV V_PAS_ < 12 cm^3^, Group B2: No RSAV V_PAS_ ≥ 12 cm^3^). SASN was found in 30 patients (mean age 30.0 ± 10.9), while 70 patients showed no narrowing (mean age 26.1 ± 7.6). Gender distribution, age, body height, body weight, and body mass index did not differ significantly between the two groups. RSAV was found in 32 patients (mean age 27.2 ± 8.7), while 68 patients displayed no reduced volume (mean age 27.3 ± 9.0). Body height was significantly higher in the non-RSAV group. Gender distribution, age, body weight, and body mass index did not differ significantly between the two groups. Table 2 shows the demographic characteristics of the patients.

### 3.2. Reproducibility

We measured the intra- and interobserver reliability and reproducibility for all of the two- and three-dimensional CPs mentioned above. Regarding the three-dimensional airway parameters, there was very good intraobserver reliability with an ICC of 0.969 [0.918–0.988] for A_min_ and 0.939 [0.848–0.976] for V_PAS_, respectively. Interobserver reliability was very good with an ICC of 0.914 [0.380–0.976] for A_min,_ and good with an ICC of 0.875 [0.297–0.963] for V_PAS_. Two-dimensional airway parameters (IPAS; MPAS; SPAS) could be reproduced with good reliability for both inter- and intraobserver reliability. The weakest value for intraobserver reliability was measured for IPAS with an ICC of 0.889 [0.702–0.957]. Regarding interobserver reliability, the weakest value was measured for MPAS with an ICC of 0.771 [0.416–0.910]. Bone-bound cephalometric landmarks in digital lateral cephalometry are known to be reliable and reproduceable [40]. In our subgroups, cephalometric parameters led to very good intraobserver and interobserver reliability. For interobserver reliability, the weakest value was measured for NSAr with an ICC of 0.931 [0.841–0.973]. For intraobserver reliability, the weakest value was measured for SArGo with an ICC of 0.941 [0.855–0.977].

### 3.3. Differences in CPs according to Three-Dimensional SASN and RSAV

Intergroup comparisons showed statistically significant differences for the two-dimensional respiratory parameters between the two SASN groups, as well as between the two RSAV groups (Table 3). In the SASN group (A_min_ < 80 mm^2^), SPAS was 9.6 ± 2.9, while it was 14.3 ± 3.1 in the group without SASN (*p* < 0.01). MPAS was 8.0 ± 2.2 in the narrowing group, while it was 12.2 ± 3.3 in the group without narrowing (*p* < 0.01). With 9.3 ± 3.0, IPAS was significantly reduced in the SASN group (A_min_ < 80 mm^2^), compared to 15.0 ± 5.1 in the group without SASN (*p* < 0.01). The three-dimensional parameters A_Sag_, V_PAS_, A_IAS_, A_MAS_, and A_SPAS_ also showed significant differences with lower dimensions in the SASN group (*p* < 0.01). Among the bone-bound parameters, the lowest *p*-value was reached for mandibular body length GoGn (*p* = 0.09) between the two groups, with 82.4 ± 7.5 in the narrowing group and 86.1 ± 7.7 in the group without narrowing without reaching statistical significance after correcting for multiple testing. Moreover, none of the cephalometric and clinical differences among the other CPs investigated regarding the dimension of the cranial base, the sagittal, the vertical or transversal dimension of he upper and lower jaw, face height, hyoid position, or the dimension of the soft palate reached statistical significance. In the RSAV group, SPAS was 10.2 ± 2.8, while it was 14.2 ± 3.5 in the group without RSAV (*p* < 0.01).

MPAS was 8.6 ± 2.6 in the reduced volume group, while it was 12.0 ± 3.5 in the group without reduced volume (*p* < 0.01). With 9.5 ± 2.9, IPAS was significantly reduced in the RSAV group when compared to 15.1 ± 5.2 in the non-RSAV group (*p* < 0.01). The three-dimensional parameters A_min_, A_Sag_, A_IAS_, A_MAS,_ and A_SPAS_ also showed significant differences with lower dimensions in the RSAV group (*p* < 0.01). In the sagittal dimension, the mandibular body length GoGn differed significantly between the two groups, with 81.7 ± 5.7 in the reduced volume group and 86.6 ± 8.2 in the group without reduced volume (*p* < 0.01). In the vertical dimension, the lower face height (Ans-Me) and overall anterior face height (N-Me) were significantly higher in the non-RSAV group (*p* < 0.05). MP-Hy was 10.2 ± 7.2 in the RSAV group, while it was 14.1 ± 6.3 in the non-RSAV group (*p* < 0.05). Among dental parameters, only OBs were significantly higher in the RASV group when compared to the non-RSAV group (*p* < 0.05). CPs regarding the dimension of the cranial base or the soft palate did not reach statistical significance.

### 3.4. Correlation Analysis of Two-Dimensional CPs to Three-Dimensional SASN

Correlation analysis revealed strong, statistically highly significant correlations between A_min_ and the two-dimensional respiratory parameters SPAS (r = 0.706; *p* < 0.01), MPAS (r = 0.713; *p* < 0.01), and IPAS (r = 0.684; *p* < 0.01), as well as the three-dimensional parameters V_PAS_ (r = 0.857; *p* < 0.01), A_Sag_ (r = 0.708; *p* < 0.01), A_SPAS_ (r = 0.812; *p* < 0.01), A_MPAS_ (r = 0.888; *p* < 0.01), and A_IAS_ (r = 0.835; *p* < 0.01; Table 4). Moreover, a strong highly significant correlation was found with GoGn (r = 0.384; *p* < 0.01). After correction for multiple testing, there was no statistically significant correlation to any other investigated CP. Regarding V_PAS_, the correlation analysis revealed strong, statistically highly significant correlations with the two-dimensional respiratory parameters SPAS (r = 0.611; *p* < 0.01), MPAS (r = 0.662; *p* < 0.01), and IPAS (r = 0.752; *p* < 0.01), as well as the three-dimensional parameters A_min_ (r = 0.857; *p* < 0.01), A_Sag_ (r = 0.873; *p* < 0.01), A_SPAS_ (r = 0.736; *p* < 0.01), A_MAS_ (r = 0.835; *p* < 0.01), and A_IAS_ (r = 0.890; *p* < 0.01). Moreover, a strong, highly significant correlation was found with S-N (r = 0.303; *p* < 0.01), GoGn (r = 0.426; *p* < 0.01), NMe (r = 0.332; *p* < 0.01), N-Ans (r = 0.309; *p* < 0.01), and Ans-Me (r = 0.310; *p* < 0.01). A medium significant correlation was found for BH (r = 0.282; *p* < 0.05), SGo (r = 0.296; *p* < 0.05), and DCH (r = 0.287; *p* < 0.05). After correction for multiple testing, there was no statistically significant correlation to any other investigated CP.

### 3.5. Multiple Linear Regression Analysis for Predicting Three-Dimensional SASN from Two-Dimensional CPs

Two-dimensional CPs associated with three dimensional SASN based on multiple linear regression analysis, with the stepwise inclusion and exclusion of all clinical and two-dimensional CPs accessible in lateral cephalometric X-rays, are shown in Table 5. SPAS (β = 0.336, *p* < 0.01), MPAS (β = 0.264, *p* < 0.05), and IPAS (β = 0.251, *p* < 0.05) were independently associated with the three-dimensional minimal cross-sectional area A_min_ (R^2^_adj_ = 0.584). IPAS (β = 0.733, *p* < 0.001) and BH (β = 0.221, *p* < 0.01) were independently associated with the superior airway space volume V_PAS_ (R^2^_adj_ = 0.606).

### 3.6. Effectiveness of Predicting Three-Dimensional SASN and RSAV from Two-Dimensional CPs

Within the ROC curve analysis, A_min_ ≤ 80 mm^2^ (n_1_ = 30; n_2_ = 70) was tested as a binary classifier against the multiple linear regression model (MLRM), predicting A_min_ as well as the individual model components, or rather the four two-dimensional CP with the strongest individual correlation to A_min_. The MLRM for A_min_ led to an AUC of 0.903 (95% confidence interval [0.841–0.966]; *p* < 0.01; Table 6). SPAS led to an AUC of 0.875 [0.795–0955] (*p* < 0.01), MPAS led to an AUC of 0.867 [0.791–0943] (*p* < 0.01), and IPAS led to an AUC of 0.858 [0.779–0937] (*p* < 0.01). GoGn led to an AUC of 0.627 [0.510–0.744] (*p* = 0.033).

V_PAS_ ≤ 12.0 cm^3^ (n_1_ = 32; n_2_ = 68) was tested as a binary classifier against the MLRM, predicting V_PAS_ as well as the individual model components, or rather the four two-dimensional CPs with the strongest individual correlations to V_PAS_. The multiple linear regression model for V_PAS_ led to an AUC of 0.881 [0.821–0.947] (*p* < 0.01). IPAS led to an AUC of 0.843 [0.767–0920] (*p* < 0.01), and BH led to an AUC of 0.683 [0.578–0787] (*p* < 0.01). SPAS led to an AUC of 0.814 [0.726–0902] (*p* < 0.01), MPAS led to an AUC of 0.804 [0.708–0900] (*p* < 0.01), and GoGn led to an AUC of 0.683 [0.578–0.787] (*p* < 0.01).

### 3.7. Definition of Cut-Off Values for the Indication of Three-Dimensional Imaging

To improve the accuracy when compared to the two-way, single cut-off classification, an intermediate range of values was defined for all parameters analyzed through ROC-curve analysis, as shown in Figure 2 and Figure 3. The lower cut-off value for three-dimensional imaging was defined as the individual value where sensitivity in the ROC-curve analysis is 90%. The upper cut-off value for three-dimensional imaging was defined as the individual value where specificity in the ROC-curve analysis is 90%, leading to an overall prevalence-independent accuracy of >90%, as shown in Table 6. Regarding A_min,_ the range of values indicating the necessity of further imaging ranges for the MLRM from 100.0 to 133.0, for SPAS from 10.4 to 12.5, for MPAS from 8.1 to 10.2, for IPAS from 9.4 to 13.5, and for GoGn from 77.0 to 90.2. For V_PAS,_ the range of values indicating the necessity of further imaging ranges for the MLRM from 13.0 to 16.4, for IPAS from 9.6 to 14.0, for MPAS from 8.1 to 13.2, for SPAS from 10.0 to 14.2, for BH from 160.0 to 179.5, and for GoGn from 74.4 to 88.8.

## 4. Discussion

A cephalometric diagnostic is highly recommended by the current guidelines regarding sleep-related breathing disorders [5,6,7]. Nevertheless, there is no clear recommendation regarding the necessary scope of such a cephalometric assessment or the indication for three-dimensional imaging [8]. This study proposes a three-dimensional cephalometric and superior airway analysis using CT in the supine position to generate two-dimensional cephalometric reconstructions, as well as three-dimensional volumetric reconstructions and volume segmentations to measure the angles, linear distances, cross-sectional areas, and volumetric measurements of the facial skeleton and the superior airway to associate traditional two-dimensional cephalometric parameters to three-dimensional cross-sectional areas and volumes. Numerous studies have been published in which CT is used to evaluate facial skeletons and upper airways, showing high reliability and reproducibility [31,41,42,43,44]. As three-dimensional SASN and RSAV is highly associated with OSAS, the further characterization of the association of SASN, RSAV, and two-dimensional CPs is necessary [37].

Traditional two-dimensional lateral cephalometric screening parameters for SASN are the sagittal airway diameter in different anatomical definitions (IPAS, MPAS, SPAS), as well as the retrognathic position of the mandible defined by the SNB-angle and the dolichofacial type defined by the NSGn-angle [27]. The proposed cut-off values are <10–11 mm for IPAS, <77° for SNB, and >67° for NSGn [26,27,28,29]. However, these definitions remain indistinctive because of their markedly overlapping value range between the affected and non-affected groups and, thus, the insufficient diagnostic selectivity [30]. Most published values were obtained with patients sitting or standing in an upright position, distinguishing the three-dimensional complexity of the anatomical conditions from the two-dimensional measurements on a single spatial plane. Airway space morphology is a dynamic condition that depends on the body position, tongue position, breathing phase, and physical tension and excitability [43,45,46,47]. Around 15% of patients who were treated with maxillomandibular advancement for OSAS do not respond as assumed, and suitable patients for surgical interventions may not be sufficiently identified [35]. Three-dimensional imaging with CT and CBCT is universally available and established in almost all clinical contexts [48]. In order to establish a clinical pathway for three-dimensional imaging as part of the screening for SASN in non-OSAS and OSAS patients, helpful definitions for indications are required based on clinical and conventional radiographic parameters.

While there is a lot of information about the association of two-dimensional cephalometric parameters with clinical conditions like OSAS, there is much less information about the association of these parameters with the three-dimensional airway morphology itself.

In our study, patients with three-dimensional SASN (A_min_ < 80 mm^2^) were found to show significantly lower values for the two-dimensional airway space parameters IPAS, MPAS, and SPAS, as well as for the mandibular body length GoGn. No significance could be reached for the other “traditional” factors like SNB and NSGn. Regarding the RSAV (V_PAS_ < 12 cm^3^), the IPAS, MPAS, and SPAS, as well as the GoGn showed significantly lower values among others associated with vertical dimensions and facial height, as well as hyoid position and dental relation. No significance could be reached for the “traditional” factors like SNB and NSGn either. SPAS, MPAS, IPAS, and GoGn could be identified as independent factors predicting A_min_ in stepwise multiple linear regression analyses. IPAS and BH were identified as independent factors predicting V_PAS_, respectively. This is in accordance with the literature, as a strong association of two-dimensional posterior airway space parameters (SPAS, MPAS, IPAS) with SASN and OSAS has been demonstrated by multiple authors [27,49,50]. A reduced mandibular body length (GoGn) is also a common factor in patients with SASN, RSAV, or OSAS, and has been confirmed in single-center studies and meta-analyses [49,50,51]. A lacking association between SNB and the posterior airway space was described in OSAS and non-OSAS patients as well [50,51,52], while there is also strong evidence associating a reduced SNB with sleep-related breathing disorders [27,49]. An association of two-dimensional radiographic parameters to clinical conditions or other two-dimensional measurements does not automatically predict the same behavior regarding three-dimensional outcome variables. This may explain the lacking association of factors regularly associated with OSAS, but not with SASN, like SNB, MP-Hy, or NSGn [27,49].

In order to perform a screening for SASN and RSAV with a distinct diagnostic benefit, both MLRM and individual factors were tested with ROC-curve analysis. For A_min,_ a balanced accuracy of 82.3% could be reached by the MLRM and from 62.9% to 83.1% by the individual factors, respectively. The cut-off values for the individual parameters are in the same range, as already mentioned in the literature [27,29]. For V_PAS,_ a balanced accuracy of 81.5% was reached for the MLRM and 68.9–77.2% for the individual factors analyzed. This seems to be a valuable contribution to the screening for SASN and RSAV indeed. Recent studies have examined the correlation of 2D lateral cephalometric measurements with 3D CT measurements of the airway. Two-dimensional measurements tend to have varying heterogeneous diagnostic impacts regarding the prediction of the 3D volume or the cross-sectional surface areas when comparing different studies [31,32,33,34]. Sears et al. state that two-dimensional lateral cephalogram cannot be considered a surrogate imaging tool to three-dimensional airway measurements [33]. The reliability and reproducibility of both methods was shown by Abramson et al.; however, they could not show a correlation between the lateral cephalometric measurements and three-dimensional measurements, except between IPAS and a retroglossal airway diameter in CT [31]. Nevertheless, Riley et al. described a strong correlation between IPAS and airway volume in CT [53]. Pirila-Parkkinen et al. compared two-dimensional cephalometric and three-dimensional MRI volume and surface measurements, finding a strong correlation in the nasopharyngeal and retropalatal regions [34]. A factor which is crucial when interpreting our results is the fact that the lateral cephalogram was reconstructed from CT data in the best possible virtual alignment of the head along the radiographic axes, and, moreover, the acquisition of both volumetric and two-dimensional cephalometric data were gained from a patient in the supine position with habitual occlusion and lips at rest. Posture-dependent changes in the superior airway reflex and gravitational force contribute to changes in the superior airway dimension. The supine position more closely mirrors the clinically relevant position while sleeping [34]. Studies completed by other research groups support this finding. Low-dose CT taken in the supine position and CBCT taken in the upright standing position among healthy subjects were compared by Van Holsbeke et al. to evaluate the changes in the airway space dimensions. The airway space dimensions were measured as significantly smaller with increasing resistance in the supine position [54]. In the supine position, gravitational forces contribute to the posterior displacement of the soft palate and tongue and, thus, to the decrease in the airway space diameter. This may have been a factor contributing to the weaker correlation between 2D and 3D results in other studies. As superior airway space morphology is highly dependent on the acquisition circumstances, results from two-dimensional cephalometry should be interpreted cautiously and not without taking acquisition circumstances into account when drawing clinical conclusions.

To increase the diagnostic accuracy, our study defined a value range for the indication of three-dimensional imaging in order to reach an accuracy of over 90% independence from the prevalence of SASN or RSAV. Especially for A_min,_ the upper cut-off values (e.g., IPAS 13.5 mm) are markedly above the binary classification cut-off (IPAS = 10.9 mm) and the reference values from the literature (IPAS = 11.0 mm). These findings may indicate the early necessity and a possible benefit of further three-dimensional imaging in patients with two-dimensional airway space measurements in the normal range of values. Studies using two-dimensional lateral cephalometry as a surrogate for corresponding three-dimensional superior airway space morphology have been published recently [55,56]. Abe-Nickler et al. also conclude that two dimensional cephalometry used to assess three-dimensional superior airway space configuration is not reliable, since there is no sufficient correlation between the posterior–anterior distances and the corresponding cross-sectional areas [56].

Our study is subject to certain strengths and limitations. One limitation is the incomprehensive definition of the anatomical boundaries and reference points for some two-dimensional and most three-dimensional measurements regarding the facial skeleton as well as superior airway morphology. It is, therefore, methodologically impossible to make comparisons, and it is very difficult to combine data to obtain reference values. The study population represents mainly young, non-obese, non-OSAS patients out of a wide spectrum of anatomic variance and airway morphologies. This may lead to the identification of screening parameters representing a wide range of patients. Radiographic airway morphology is highly dependent on the body position. We therefore investigated and compared both three-dimensional and two-dimensional parameters in a standardized supine position, which is rare in the literature and led to a better correlation of two- and three-dimensional factors than described in most of the recent literature, implicating that acquisition circumstances are crucial in the objectification and interpretation of both two- and three-dimensional airway dimensions.

To the best of our knowledge, this study is the first to examine the correlation of a wide range of two-dimensional and three-dimensional cephalometric parameters regarding SASN from a corresponding CT dataset in the supine position, defining the cut-off values for the indication of three-dimensional imaging. It may contribute to the cephalometric assessment regarding SASN in OSAS and non-OSAS patients, and it encourages clinicians to perform three-dimensional imaging for the evaluation of SASN based on scientific recommendations. Moreover, the investigation of the clinical responses to different therapeutic approaches in the treatment of sleep-related breathing disorders with correlations to the revealed factors should be expanded.

## 5. Conclusions

Two-dimensional cephalometry is a useful initial screening method in the identification of superior airway space narrowing. Nevertheless, as the correlation of two-dimensional cephalometric parameters with three-dimensional upper airway space narrowing is varying and highly dependent on acquisition conditions, the indication for three-dimensional imaging to evaluate upper airway space morphology should be provided generously in patients with low but normal airway space parameters in two-dimensional cephalometry.

## Figures and Tables

**Figure 1 jcm-13-02685-f001:**
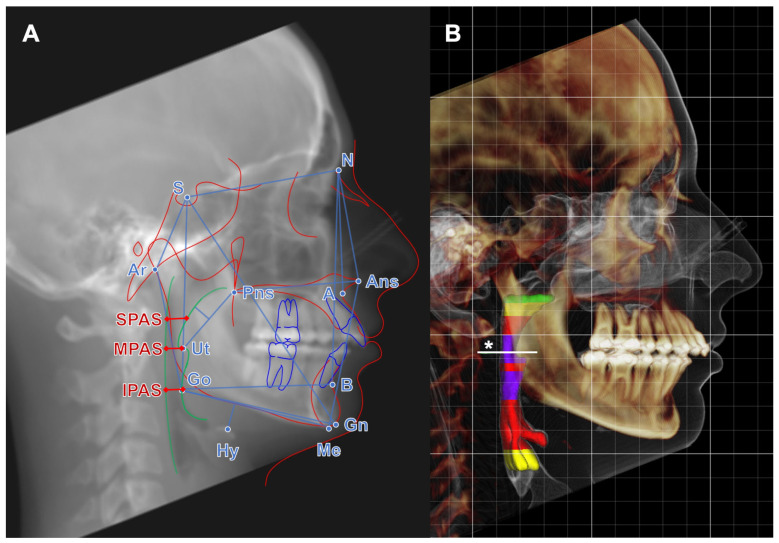
(**A**): Reconstruction of the lateral cephalometric X-ray from the CT scan. Cephalometric reference points: S (Sella), N (Nasion), A (point A), B (point B), Gn (Gnathion), Me (Menton), Hy (Hyoid), Go (Gonion), Ut (Uvula tip), Ar (Articulare), Ans (anterior nasal spine), Pns (posterior nasal spine). Airway reference diameters: SPAS (sagittal airway diameter in the middle between Uvula tip and posterior nasal spine parallel to B-Go Line), MPAS (sagittal airway diameter at Uvula tip parallel to B-Go Line), IPAS (sagittal airway diameter in B-Go Line). (**B**): Three-dimensional volume rendering from the CT scan with the segmentation of the superior airway space and the color coding of the cross-sectional area (purple: <100 mm^2^, red: 100–<200 mm^2^, yellow 200–<300 mm^2^, green ≥ 300 mm^2^, * minimal cross-sectional area).

**Figure 2 jcm-13-02685-f002:**
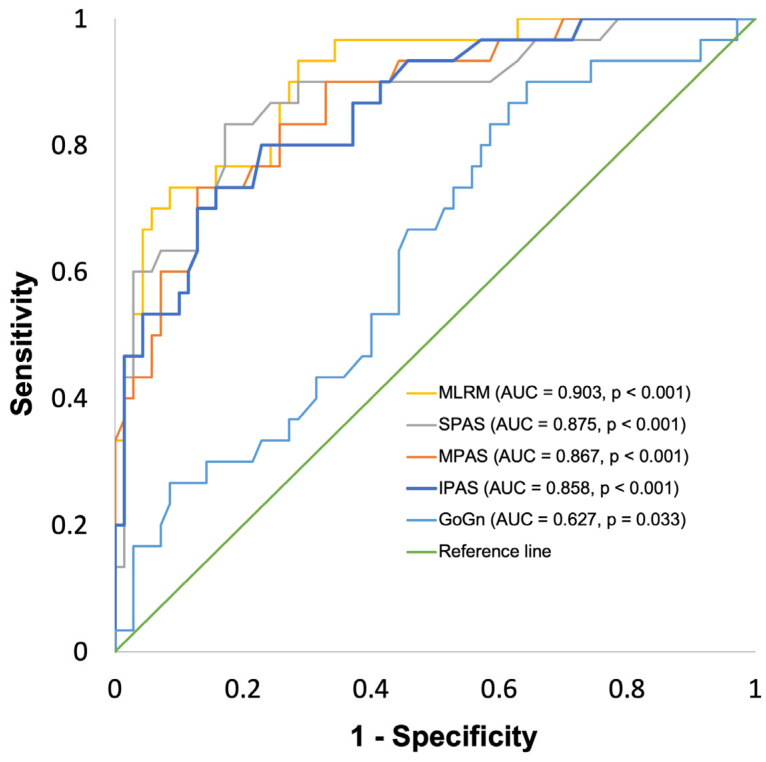
Receiver operating characteristic (ROC) curve for two-dimensional cephalometric parameters predicting A_min_. MLRM: multiple linear regression model, AUC: area under the curve.

**Figure 3 jcm-13-02685-f003:**
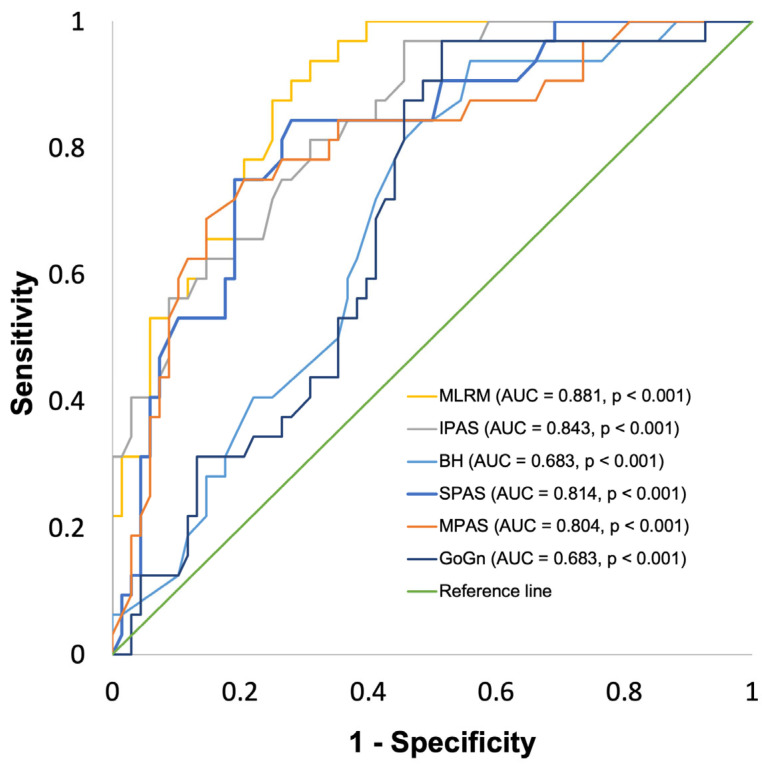
Receiver operating characteristic (ROC) curve for two-dimensional cephalometric parameters predicting V_PAS_. MLRM: multiple linear regression model, AUC: area under the curve.

**Table 1 jcm-13-02685-t001:** Definition of the measured cephalometric parameters.

	Parameter	Definition
Two-dimensional airway parameter	IPAS (mm)	Sagittal airway diameter in the B (point B)–Go (Gonion) Line
	MPAS (mm)	Sagittal airway diameter at Ut (Uvula tip) parallel to the B-Go Line
	SPAS (mm)	Sagittal airway diameter in the middle between Ut and Pns (posterior nasal spine) parallel to the B-Go Line
Three-dimensional airway parameter	A_min_ (mm^2^)	Minimal transversal airway cross-section area
	V_PAS_ (cm^3^)	Volume of posterior airway space between posterior nasal spine and hyoid plane
	A_Sag_ (mm^2^)	Maximal airway cross-section area in sagittal plane between posterior nasal spine and hyoid plane
	A_IAS_ (mm^2^)	Transversal airway cross-section area in IPAS–Plane
	A_MAS_ (mm^2^)	Transversal airway cross-section area in MPAS–Plane
	A_SPAS_ (mm^2^)	Transversal airway cross-section area in SPAS–Plane
Cranial base	S-N (mm)	Distance between S (Sella) and N (Nasion)
	NSAr (°)	Angle from N to S to Ar (Articulare)
Sagittal dimensions	SNA (°)	Angle from S to N to A (point A)
	SNB (°)	Angle from S to N to B
	ANB (°)	Angle from A to N to B
	Ans-Pns (mm)	Distance between Ans (anterior nasal spine) and Pns
	GoGn (mm)	Distance between outer Go and outer Gn (Gnathion)
Vertical dimensions	SP-SN (°)	Angle from Line Ans-Pns to Line S-N
	MP-SN (°)	Angle from Line Go-Me (Menton) to Line S-N
	SP-MP (°)	Angle from Line Ans-Pns to Line Go-Me
	SArGo (°)	Angle from N to Ar to Go
	ArGoMe (°)	Angle from Ar to Go to Me
	Sum (°)	Sum of the Angles NSAr, SArGo, and ArGoMe
	NSGn (°)	Angle from N to S to Gn (Y-Axis)
Face height	S-Go (mm)	Distance between S and Go (posterior face height)
	N-Me (mm)	Distance between N and Me (anterior face height)
	N-Ans (mm)	Distance between N and Ans (upper anterior face height)
	Ans-Me (mm)	Distance between Ans and Me (lower anterior face height)
	S-Go/N-Me	Relation of anterior to posterior face height
	Ans-Go/N-Ans	Relation of lower anterior to upper anterior face height
Hyoid position	MP-Hy (mm)	Distance from Hy (Hyoid) to Go-Me Line
Dental relations	OJ (mm)	Overjet
	OB (mm)	Overbite
Transversal dimensions	DI6 (mm)	Transversal distance between first lower molar
	DI4 (mm)	Transversal distance between first lower premolar
	DMA (mm)	Transversal mandibular distance between the mandibular angles
	DCH (mm)	Transversal mandibular distance between lateral poles of the condylar heads
Palate	SPL (mm)	Length of soft palate
	SPT (mm)	Thickness of soft palate

**Table 2 jcm-13-02685-t002:** Patient characteristics of the patients included in the analysis (n = 100).

	Group A1: SASN (A_min_ < 80 mm^2^)	Group A2: No SASN (A_min_ ≥ 80 mm^2^)	*p*-Value (A1 vs. A2)	Group B1: RSAV (V_PAS_ < 12 cm^3^)	Group B2: No RSAV (V_PAS_ ≥ 12 cm^3^)	*p*-Value (B1 vs. B2)	All Patients
n	30	70		32	68		100
Gender							
Male	13	34		9	38		47
Female	17	36		23	30		53
Gender distribution m/f	43.3%/ 56.7%	48.6%/ 51.4%	0.98 °	28.1%/ 71.9%	55.9%/ 44.1%	0.98 °	47.0%/ 53.0%
Age	30.0 ± 10.9	26.1 ± 7.6	0.20 *	27.2 ± 8.7	27.3 ± 9.0	0.97 *	27.2 ± 8.9
Body characteristics							
Body height BH (cm)	171.2	175.2	0.44 *	169.8	175.9	<0.01 *	174.0
Body weight BW (kg)	75.8	76.6	0.26 *	72.2	78.4	0.26 *	76.4
Body mass index BMI	25.7	24.8	0.16 *	24.9	25.1	0.96 *	25.0

* student’s *t*-test; ° chi-squared test.

**Table 3 jcm-13-02685-t003:** Intergroup comparison (n = 100).

Parameter		Group A1: SASN (A_min_ ≤ 80 mm^2^)	Group A2: No SASN (A_min_ > 80 mm^2^)	*p*-Value (A1 vs. A2)	Group B1: RASV (V_PAS_ < 12 cm^3^)	Group B2: No RASV (V_PAS_ ≥ 12 cm^3^)	*p*-Value (B1 vs. B2)	All Patients
	n=	30	70		32	68		100
Three- dimensional airway parameter	A_min_	55.9 ± 18.1	180.3 ± 82.6	-	72.1 ± 31.3	176.3 ± 89.7	<0.01 *	143.0 ± 90.2
	V_PAS_	11.08 ± 3.00	19.85 ± 8.08	<0.01 *	9.75 ± 1.60	20.72 ± 7.40	-	17.2 ± 8.0
	A_Sag_	614.6 ± 161.1	873.4 ± 234.0	<0.01 *	577.3 ± 137.3	898.5 ± 215.4	<0.01 *	795.7 ± 244.9
	A_IAS_	133.0 ± 59.9	312.0 ± 159.8	<0.01 *	133.9 ± 45.0	316.9 ± 161.4	<0.01 *	242.7 ± 153.6
	A_MAS_	90.7 ± 46.6	242.7 ± 126.3	<0.01 *	100.3 ± 42.2	242.7 ± 131.1	<0.01 *	201.4 ± 137.3
	A_SPAS_	163.0 ± 63.5	283.2 ± 102.1	<0.01 *	164.4 ± 53.	286.1 ± 104.4	<0.01 *	337.9 ± 106.5
Two-dimensional airway parameter	SPAS	9.5 ± 2.9	14.3 ± 3.1	<0.01 *	10.2 ± 2.8	14.2 ± 3.5	<0.01 *	12.9 ± 3.7
	MPAS	8.0 ± 2.2	12.2 ± 3.3	<0.01 *	8.6 ± 2.6	12.0 ± 3.5	<0.01 *	10.9 ± 3.6
	IPAS	9.3 ± 3.0	15.0 ± 5.1	<0.01 *	9.5 ± 2.9	15.1 ± 5.2	<0.01 *	13.3 ± 5.3
Cranial base	S-N	74.3 ± 5.3	75.8 ± 5.0	0.38	73.9 ± 5.4	76.0 ± 4.9	0.16	75.3 ± 5.1
	NSAr	124.2 ± 5.6	123.2 ± 6.8	0.61	124.9 ± 6.3	122.8 ± 6.5	0.28	123.5 ± 5.5
Sagittal dimensions	SNA	79.8 ± 3.6	80.8 ± 4.9	0.46	80.2 ± 4.	80.6 ± 4.8	0.78	80.5 ± 4.6
	SNB	76.7 ± 5.6	78.5 ± 5.8	0.29	76.7 ± 4.9	80.6 ± 4.8	0.25	77.9 ± 5.8
	ANB	3.1 ± 5.1	2.3 ± 5.0	0.62	3.5 ± 4.1	2.1 ± 5.3	0.31	2.6 ± 5.0
	Ans-Pns	55.9 ± 4.3	56.5 ± 5.3	0.72	55.4 ± 5.0	56.8 ± 5.0	0.35	56.4 ± 5.0
	Go-Gn	82.4 ± 7.5	86.1 ± 7.7	0.09	81.7 ± 5.7	86.6 ± 8.2	<0.01 *	85.0 ± 7.8
Vertical dimensions	SP-SN	8.2 ± 3.6	8.1 ± 3.6	0.95	8.1 ± 3.7	8.1 ± 3.6	0.98	8.1 ± 3.6
	MP-SN	37.1 ± 9.0	35.3 ± 7.0	0.49	34.6 ± 7.5	36.4 ± 7.7	0.45	35.8 ± 7.6
	SP-MP	28.9 ± 9.0	27.2 ± 6.7	0.53	26.5 ± 8.2	28.2 ± 7.	0.48	27.7 ± 7.4
	SArGo	145.2 ± 7.1	145.4 ± 8.8	0.97	145.4 ± 8.3	145.3 ± 8.3	0.98	145.4 ± 8.3
	ArGoMe	127.7 ± 10.3	126.6 ± 8.3	0.78	124.3 ± 8.9	128.2 ± 8.7	0.13	127.0 ± 8.9
	Sum	397.1 ± 9.0	395.3 ± 7.0	0.49	394.6 ± 7.5	396.4 ± 7.7	0.46	395.8 ± 7.6
	NSGn	71.0 ± 5.4	68.8 ± 5.2	0.19	69.8 ± 4.6	69.3 ± 5.7	0.79	69.5 ± 5.4
Face height	S-Go	85.5 ± 10.2	86.2 ± 8.4	0.89	83.9 ± 8.7	87. ± 8.9	0.25	86.0 ± 8.9
	N-Me	131.7 ± 12.2	132.3 ± 11.9	0.94	127.4 ± 11.8	134.3 ± 11.4	0.03 *	132.1 ± 11.9
	N-Ans	56.1 ± 4.5	56.6 ± 4.4	0.78	55.0 ± 4.5	57.1 ± 4.3	0.11	56.4 ± 4.4
	Ans-Me	73.6 ± 9.7	74.3 ± 8.8	0.88	70.5 ± 9.2	75.7 ± 8.5	0.03 *	74.1 ± 9.0
	S-Go/ N-Me	0.65 ± 0.06	0.65 ± 0.05	0.95	0.66 ± 0.06	0.65 ± 0.05	0.49	65.3 ± 5.5
	Ans-Me/ N-Ans	1.31 ± 0.14	1.32 ± 0.17	0.98	1.29 ± 0.17	1.33 ± 0.14	0.38	1.32 ± 0.15
Hyoid position	MP-Hy	12.7 ± 6.8	13.2 ± 7.1	0.87	10.2 ± 7.2	14.1 ± 6.3	0.04 *	12.9 ± 6.8
Dental relations	OJ	1.6 ± 5.8	2.5 ± 5.8	0.62	3.7 ± 5.0	1.0 ± 5.9	0.07	1.9 ± 5.8
	OB	−0.2 ± 3.3	1.0 ± 2.5	0.14	1.3 ± 2.6	−0.4 ± 3.2	0.03 *	0.2 ± 3.1
Transversal dimensions	DI4	36.3 ± 2.5	36.3 ± 2.3	0.98	35.5 ± 2.1	36.6 ± 2.5	0.13	47.6 ± 3.6
	DI6	47.5 ± 3.8	48.1 ± 3.1	0.61	47.2 ± 2.9	47.8 ± 3.9	0.62	36.3 ± 2.4
	DMA	104.8 ± 8.7	104.5 ± 9.0	0.97	102.6 ± 9.4	105.8 ± 8.3	0.26	104.8 ± 8.7
	DCH	115.3 ± 7.4	116.6 ± 7.1	0.56	114.2 ± 5.9	116.4 ± 7.8	0.27	115.7 ± 7.3
Palate	SPL	36.6 ± 4.9	37.8 ± 4.9	0.45	36.1 ± 5.0	37.3 ± 4.9	0.45	36.9 ± 4.9
	SPT	7.3 ± 2.1	8.1 ± 2.4	0.28	7.5 ± 1.8	7.6 ± 2.4	0.96	7.6 ± 2.2

* student’s *t*-test *p* < 0.05.

**Table 4 jcm-13-02685-t004:** Pearson correlation r_p_ between cephalometric parameters, minimal cross-sectional area of the superior airway space (A_min_), and superior airway space volume (V_PAS_) (n = 100).

		A_min_		V_PAS_	
		r_p_	*p*-Value	r_p_	*p*-Value
Patient Characteristics	Age	−0.169	0.22	−0.005	1.00
	BH	0.015	0.26	0.282	0.02 *
	BW	0.018	0.99	0.187	0.17
	BMI	−0.088	0.56	0.063	0.72
Three- dimensional airway parameter	A_min_	-	-	0.857	<0.01 *
	V_PAS_	0.857	<0.01 *	-	-
	A_sag_	0.708	<0.01 *	0.873	<0.01 *
	A_SPAS_	0.812	<0.01 *	0.736	<0.01 *
	A_MAS_	0.888	<0.01 *	0.835	<0.01 *
	A_IAS_	0.835	<0.01 *	0.890	<0.01 *
Two-dimensional airway parameter	SPAS	0.706	<0.01 *	0.611	<0.01 *
	MPAS	0.713	<0.01 *	0.662	<0.01 *
	IPAS	0.684	<0.01 *	0.752	<0.01 *
Cranial base	S-N	0.208	0.11	0.303	<0.01 *
	NSAr	−0.094	0.53	−0.118	0.43
Sagittal dimensions	SNA	0.148	0.30	0.083	0.59
	SNB	0.211	0.11	0.183	0.17
	ANB	−0.109	0.45	−0.136	0.36
	Ans-Pns	0.110	0.46	0.229	0.07
	Go-Gn	0.384	<0.01 *	0.426	<0.01 *
Vertical dimensions	SP-SN	−0.018	0.96	−0.020	1.00
	MP-SN	−0.109	0.45	−0.018	0.95
	SP-MP	−0.104	0.46	−0.008	0.99
	SArGo	0.019	1.00	0.019	1.00
	ArGoMe	−0.043	0.86	0.053	0.80
	Sum	−0.110	0.45	−0.018	0.98
	NSGn	−0.218	0.09	−0.113	0.45
Face height	S-Go (PFH)	0.121	0.42	0.296	0.01 *
	N-Me (AFH)	0.121	0.43	0.332	<0.01 *
	N-Ans	0.141	0.33	0.309	<0.01 *
	Ans-Me	0.129	0.39	0.310	<0.01 *
	S-Go/N-Me	0.014	0.96	0.007	0.99
	Ans-Me/N-Ans	0.036	0.90	0.115	0.45
Hyoid position	MP-Hy	−0.052	0.80	0.178	0.19
Dental relations	OJ	−0.154	0.28	−0.184	0.18
	OB	0.151	0.29	0.210	0.11
Transversal dimensions	DI4	−0.030	0.95	0.080	0.64
	DI6	0.048	0.85	0.179	0.26
	DMA	0.074	0.65	0.223	0.08
	DCH	0.014	0.47	0.287	0.01 *
Palate	SPL	−0.123	0.42	0.015	0.96
	SPT	−0.093	0.53	0.035	0.90

* *p* < 0.05.

**Table 5 jcm-13-02685-t005:** Multiple linear regression models predicting A_min_ and V_PAS_ from the two-dimensional cephalometric assessment.

Dependent Variable	Independent Variable	Regression Coefficient B	Standard Error SE	Standardized Coefficient ß	T	R^2^	R^2^_adj_	F
A_min_	Constant	−91.215 ***	21.382	-	−4.266	0.597	0.584	47.325 ***
	SPAS	8.126 **	2.464	0.336 **	3.297			(df = 3)
	MPAS	6.636 *	2.982	0.264 *	2.226			
	IPAS	4.290 *	1.803	0.251 *	2.379			
V_PAS_	Constant	−29.904 **	9.263	-		0.614	0.606	77.019 ***
	IPAS	1.117 ***	0.096	0.733 ***	11.579			(df = 2)
	BH	0.185 **	0.053	0.221 ***	3.482			

* *p* < 0.05; ** *p* < 0.01; *** *p* < 0.001.

**Table 6 jcm-13-02685-t006:** Receiver operating characteristic (ROC) curve analysis for two-dimensional cephalometric parameters predicting A_min_ and V_PAS_. Definition of the lower and upper cut-off values for an indication of three-dimensional imaging.

Dependent Variable	Predictor	AUC [95% CI]	Cut-Off	Sensitivity (%) [95% CI]	Specificity (%) [95% CI]	Balanced Accuracy (%)	Lower Cut-Off	Upper Cut-Off	Balanced Accuracy (%)
A_min_	MLRM A_min_	0.903 ** [0.841–0.966]	100	73.3	91.4	82.3	100.0	133.0	90.7
	SPAS	0.875 ** [0.795–0.955]	11.8	0.833	0.829	83.1	10.4	12.5	90.7
	MPAS	0.867 ** [0.791–0.943]	8.8	0.733	0.871	80.2	8.1	10.2	90.0
	IPAS	0.858 ** [0.779–0.937]	10.9	0.733	0.843	78.8	9.4	13.5	90.0
	GoGn	0.627 * [0.510–0.744]	90.2	0.900	0.357	62.9	77.0	90.2	90.0
V_PAS_	MLRM V_PAS_	0.881 ** [0.817–0.947]	16.40	0.938	0.691	81.5	13.00	16.4	91.9
	IPAS	0.843 ** [0.767–0.920]	14.0	0.969	0.544	75.7	9.6	14.0	93.5
	BH	0.683 ** [0.578–0.788]	179.0	0.938	0.441	68.9	160.0	179.5	92.5
	SPAS	0.814 ** [0.726–0.902]	12.4	0.844	0.720	78.2	10.0	14.2	90.9
	MPAS	0.804 ** [0.708–0.900]	9.2	0.750	0.794	77.2	8.1	13.2	90.9
	GoGn	0.683 ** [0.578–0.787]	88.8	0.969	0.485	72.7	74.4	88.8	93.5

* *p* < 0.05; ** *p* < 0.01.

## Data Availability

The data presented in this study are available on request from the corresponding author.

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
