# Peer review of "Cephalometric Screening Assessment for Superior Airway Space Narrowing—Added Value of Three-Dimensional Imaging"

_jcm, 2024, doi:10.3390/jcm13092685_

Round 1
Reviewer 1 Report
Comments and Suggestions for Authors
This study on using cephalometric screening to detect narrowing in the upper airway marks a major step forward in diagnosing conditions like obstructive sleep apnea and planning orthognathic surgery. The suggestion to prioritize 3D imaging, especially for patients with seemingly normal airways in 2D scans, demonstrates a proactive approach to ensure accurate diagnosis and treatment planning. This research significantly improves diagnostic methods for upper airway narrowing, ultimately leading to better patient care. Overall, the comprehensive nature of this research, coupled with its valuable insights and practical recommendations.
Author Response
Thank you for your detailed and helpful advice on our manuscript “Cephalometric screening assessment for superior airway space narrowing – added value of 3-dimensional imaging”. Let's comment on this point by point:
Reviewer 1:
- We have no comments regarding this review.

Reviewer 2 Report
Comments and Suggestions for Authors
The authors presented a study to correlate the 2D to the 3D airway as shown on CT scan.
I have the following comments:
1. I found very low clinical relevance of this study. There is minimal description of the knowledge gap and clinical relevance of how this study is justified.
2. It is known that CT scan is not a suitable assessment for patients with OSA. Patients who take CT scans are awake, while OSA is a sleeping disorder. The posturing and tongue positioning all affects the images of the airway. While the authors tried to correlate the study with the need of airway analysis in OSA, it was not sufficiently explained. The included individuals were not OSA patients either.
3. The authors found that sagittal airway space and mandibular length had the highest correlation of airway space narrowing. These are known facts, if not common sense, and were well reported in the literature for years.
4. I would suggest the authors to conduct this research in OSA patients, and to correlate the findings with sleep endoscopy and their sleep parameters e.g. AHI or LSat. This would bring more clinical relevance in this area.
Author Response
Thank you for your detailed and helpful advice on our manuscript “Cephalometric screening assessment for superior airway space narrowing – added value of 3-dimensional imaging”. Let's comment on this point by point:
Reviewer 2:
- Clinical relevance: Assessing the morphology of the superior airway space is a crucial diagnostic step in the surgical treatment planning of patients with obstructive sleep apnea, prior to combined orthodontical-orthognathic surgical treatment as well as in aesthetic orthognathic surgery [1-4]. Superior airway space narrowing is along with obesity impaired muscle responsiveness leading to static or dynamic tissue laxity one important pillar in the pathogenesis of OSAS. From the perspective of an oral and maxillofacial surgeon, narrowing of the superior airway space is an important factor that can be addressed in the context of orthognathic surgery, whether for therapeutic or prophylactic indications and is therefore highly relevant from a clinician’s perspective. This was clarified within the introduction.
- CT or lateral cephalometric x-ray are not necessarily needed for or after the initial diagnosis of OSAS [5]. This was clarified within the introduction. As OSAS is a disease of multifactorial origin it is highly important to select the best possible therapy option in patients not suitable for PAP-therapy. Here it is relevant from a surgeon’s perspective to get insight into surgically addressable objectives to recommend the optimal alternative therapy. Orthognathic surgery for OSAS does make more sense in patients with superior airway narrowing as primary pathogenetic problem while hypoglossal nerve stimulation might be better in patients with increased collapsibility but normal anatomy. Moreover, surgically addressable objectives are needed for prophylactic approaches in otherwise indicated orthognathic surgeries.
- It’s true that these factors are described in the literature and might be common sense. But this work is not about how to identify them, it’s how to use them in order to deal with individual patients in consultation and treatment planning especially when more than one therapy option is left. We show the diagnostic value of a single parameters in lateral cephalometry might be weak and therefore this method alone might not be suitable for decision making in consultation and treatment planning.
- As described in point 1 clinical relevance is given not only in patients with OSAS but in all patients possibly requiring orthognathic surgery. The rationale for studying this in young adult, non-OSAS, non-obese patients is to obtain a correlation of cephalometric parameters with airway morphology that is as undisturbed as possible and not distorted by OSAS-associated cofactors such as obesity or increased collapsibility.

Reviewer 3 Report
Comments and Suggestions for Authors
Dear Authors,
Thank You for a pleasure to read Your work.
I have several offers and comments to improve Your article.
Abstract
Please, write in the beginning the whole phrase for ‘ non-OSAS’.
Please, write in the methods section used statistics criteria.
Please, check keywords according to MeSH.
Introduction
It is too short. Please, add information about outcomes of sleep apnea, possible predictors, diagnostics, and other methods of treatment with references.
M&Ms
Inclusion criteria
‘non-OSAS patients aged 18 years or older’
I consider it as the great selection bias of Your research as the age 18 is too small due to the continuous growth of skeleton and, privately, the skull.
You wrote ‘various skeletal anomalies of the jaw/skull base ratio 65 prior to orthognathic surgery between January 2016 and December 2021’ . Please, write here or in the results the accurate types of anomalies, their percent.
And the important part that You needed to count these anomalies as factor for Yous study. It is one more bias.
2.2 Please, write the slice thickness.
Line 108: ‘Unpaired t-test and Chi-squared-test were used for intergroup comparison.’
Chi-squared test is usually used for odd ration. What exactly did You count inside one group?
Lines 110-112: the strength of correlation for Pearson coefficient ia rather discussable. For example, there is a little other statement in more ‘fresh’ articles, for example, Correlation Coefficients: Appropriate Use and Interpretation Schober, Patrick MD, PhD, MMedStat; Boer, Christa PhD, MSc; Schwarte, Lothar A. MD, PhD, MBA. Anesthesia & Analgesia 126(5):p 1763-1768, May 2018. | DOI: 10.1213/ANE.0000000000002864.
The correlation coefficient must be at least not less than 0,4.
Results
Lines 145-146: Interobserver reliability was very 145 good with an ICC of 0.914 [0.380 – 0.976] for Amin and good with an ICC of 0.875 [0.297 – 146 0.963] for VPAS.
There are rather strange confidence intervals for ICC. How could You explain these data?
Table 3. You have my probabilities they must be corrected according one of correction coefficients as Bonferroni of FDR. Please, add this information and re-count results. Also, for table 4
Also, add the medians and Q1-Q3 for abnormal distributions (table 3).
3.5 Lines 219-224: ‘SPAS (β = 219 0.336, p < 0.01), MPAS (β = 0.264, p < 0.05) and IPAS (β= 0.251, p < 0.05) were independently 220 associated with three-dimensional minimal cross-sectional area Amin (R2adj = 0.584). IPAS (β 221 = 0.733, p < 0.001) and BH ( = 0.221, p < 0.01) were independently associated with superior 222 airway space volume VPAS (R2adj=0.606).’
Do You mean the power? Then it is too small. It must be at least 80%.
Sincerely, Reviewer
Author Response
Thank you for your detailed and helpful advice on our manuscript “Cephalometric screening assessment for superior airway space narrowing – added value of 3-dimensional imaging”. Let's comment on this point by point:
Reviewer 3:
- Abstract: Whole phrase for OSAS was added to the methods section. Grouping criteria, statistical criteria and methods were added to the methods section. Keyword were adjusted according to the MeSH-terms.
- Introduction: Information on the associations of sleep apnea, possible predictors, diagnostic procedures and interdisciplinary treatment methods were added with references. The importance of focusing on airway morphology in the field of orthognathic surgery in patients without phenotypically diagnosable OSAS was clarified.
- Materials & Methods – inclusion criteria “non-OSAS patients aged 18 years or older”:
Orthognathic surgery is most often performed in younger adults. Different Studies suggest a mean age about 30 years, which is in alignment with our patient group. Orthognathic surgery is considered save in regard of growth-related relapse after the age of 18 years [6]. Bondevik et al. suggest that most of the age associated cephalometric changes in adulthood are small. For the most cephalometric parameters they are within 1 mm what is in the range of the slice thickness in CT-scan [7].
- Materials & Methods – inclusion criteria “various skeletal anomalies of the jaw/skull base ratio":
Orthognathic surgery is most often indicated to correct dentofacial deformities including malocclusions owing to skeletal problems including severe class II or class III problems, anterior open bites, increased overbites and facial asymmetries and combinations of these in a continuous spectrum. These different indications are represented by the different cephalometric parameters with mean and standard deviation. The overall mean of ANB as the most important factor describing sagittal discrepancy for example is 2.6° which is within its reference value for class I occlusion (2°±2°) with a broad standard deviation of 5.0° indicating the inclusion of patients with class II and class III sagittal discrepancy. As dentofacial deformities are most often a combination of different anomalies in different severities, there is in our opinion no added value in providing classified classifications alongside the continuous spectrum of cephalometric parameters. The continuous nature of dentofacial discrepancies was highlighted in the text.
- Materials & Methods – slice Thickness: Slice thickness of 1 mm was added to the M&M section.
- Materials & Methods “Unpaired t-test and Chi-squared-test were used for intergroup comparison”: Chi-squared was used for gender ratio comparison between the groups. This was clarified within the text.
- Materials & Methods “ the strength of correlation”:
The strength of correlation and interpretation of correlation coefficients is a crucial step in interpreting scientific data. We do not use the cited work by Hemphil et al. as a final statement of data interpretation[8]. It is only used as a verbal translation of the data obtained, but no final interpretation is derived from it. Schober et al. also suggest embedding the results obtained in a larger context and warn against an unreflected interpretation based on fixed values [9]. We also do this with the further processing of the data via various other methods. - Results “intraclass correlation coefficient”:
Confidence intervals of Intraclass correlation are dependent from the number of raters. In test-retest situations with only 2 raters and only a distinct number of samples CI can usually be wide which is often not reported. ICC formulation was added to the M&M section[10]. - Results “multiple test correction”: p-values were adjusted for multiple testing by FDR-Adjustment according to Benjamini & Hochberg [11].
- Results “standardized coefficient ß”: ß does not represent power but the standardized coefficient in multiple linear regression according to table 5.

Reviewer 4 Report
Comments and Suggestions for Authors
The authors formally and statistically accurately describe and analyse the possible contribution of 3D imaging in the diagnosis of sleep disorders.
Author Response
Thank you for your detailed and helpful advice on our manuscript “Cephalometric screening assessment for superior airway space narrowing – added value of 3-dimensional imaging”. Let's comment on this point by point:
Reviewer 4:
- We have no comments regarding this review.

Round 2
Reviewer 2 Report
Comments and Suggestions for Authors
The authors did not address my concerns. I am not able to support acceptance of this work
Comments on the Quality of English LanguageCan be improved
Reviewer 3 Report
Comments and Suggestions for Authors
Dear Authors,
Thank You for Your corrections.
Sincerely, Reviewer